# Elevated Procalcitonin Is Positively Associated with the Severity of COVID-19: A Meta-Analysis Based on 10 Cohort Studies

**DOI:** 10.3390/medicina57060594

**Published:** 2021-06-09

**Authors:** Yue Shen, Cheng Cheng, Xue Zheng, Yuefei Jin, Guangcai Duan, Mengshi Chen, Shuaiyin Chen

**Affiliations:** 1Department of Epidemiology and Health Statistics, College of Public Health, Zhengzhou University, Zhengzhou 450000, China; 202022272014924@gs.zzu.edu.cn (Y.S.); chengc2019@gs.zzu.edu.cn (C.C.); 202022272014971@gs.zzu.edu.cn (X.Z.); jyf201907@zzu.edu.cn (Y.J.); gcduan@zzu.edu.cn (G.D.); 2Hunan Provincial Key Laboratory of Clinical Epidemiology, Xiangya School of Public Health, Central South University, Changsha 410078, China; chenmengshi@csu.edu.cn

**Keywords:** COVID-19, SARS-CoV-2, procalcitonin, severity, meta-analysis

## Abstract

*Background and Objectives*: Procalcitonin (PCT) is positively associated with the severity of COVID-19 (including severe, critical, or fatal outcomes), but some of the confounding factors are not considered. The aim of this meta-analysis was to estimate the adjusted relationship between elevated procalcitonin on admission and the severity of COVID-19. *Materials and Methods:* We searched 1805 articles from PubMed, Web of Science, and Embase databases up to 2 April 2021. The articles were selected which reported the adjusted relationship applying multivariate analysis between PCT and the severity of COVID-19. The pooled effect estimate was calculated by the random-effects model. *Results:* The meta-analysis included 10 cohort studies with a total of 7716 patients. Patients with elevated procalcitonin on admission were at a higher risk of severe and critical COVID-19 (pooled effect estimate: 1.77, 95% confidence interval (CI): 1.38–2.29; *I*^2^ = 85.6%, *p* < 0.001). Similar results were also observed in dead patients (pooled effect estimate: 1.77, 95% CI: 1.36–2.30). After adjusting for diabetes, the positive association between PCT and the severity of COVID-19 decreased. Subgroup analysis revealed heterogeneity between studies and sensitivity analysis showed that the results were robust. There was no evidence of publication bias by Egger’s test (*p* = 0.106). *Conclusions:* Higher procalcitonin is positively associated with the severity of COVID-19, which is a potential biomarker to evaluate the severity of COVID-19 and predict the prognosis.

## 1. Introduction

The Coronavirus Disease 2019 (COVID-19) pandemic has spread rapidly around the world in a short time, resulting in more than 116 million cases worldwide and 2.62 million deaths [1]. Of the 21.2 million existing cases, more than 89,000 were severe or critical COVID-19, which posed a considerable threat to global health and economies.

Patients with severe COVID-19 often have high levels of inflammatory factors in serum [2]. Inflammatory cytokine storm plays an important role in the development of COVID-19 patients [3]. SARS-CoV-2, the pathogen of COVID-19, can trigger immune response against the infection. However, an excessive immune response will trigger a cytokine storm, which will cause a series of bad effects on the body [4]. In addition, tocilizumab, an IL-6 receptor blocker, neutralizes IL-6 activity by binding the IL-6R at the IL-6-binding epitope. The preliminary results from the RECOVERY trial identified a slight decrease in 28-day mortality in patients treated with the combination of steroids and tocilizumab versus steroids alone. Furthermore, mortality was 4–5% higher in patients who received tocilizumab monotherapy compared with those who did not receive tocilizumab or steroids [5]. Therefore, inflammatory biomarkers, such as PCT, C-reactive protein, and interleukin (IL)-6, may be potential targets for the treatment or prediction of COVID-19 [6].

PCT is a biomarker of systemic inflammatory activity in the early phase after infection resulting from pro-inflammatory stimuli [7], which is bound up with the prognosis of infectious diseases. For instance, PCT can be used as an independent risk factor affecting the prognosis of patients with sepsis (OR = 1.087, 95% CI: 1.022–1.157) [8]. Furthermore, studies have reported that PCT is associated with the severity of COVID-19 [9,10,11]. A retrospective study suggested that PCT was a risk factor of in-hospital death from COVID-19 (OR = 6.350, 95% CI: 1.396–28.882) [11]. However, it was worth noting that PCT was considered as an important risk factor of the severity of COVID-19 based on univariate analysis (OR = 1.13, 95% CI: 1.03–1.24), which is inconsistent with those based on multivariate analysis after adjusting for confounding factors (OR = 1.05, 95% CI: 0.96–1.15) [12]. It suggested that the association between PCT and the severity of COVID-19 might be confounded by some confounding factors. More studies using multivariate analysis adjusting for confounding factors with inconsistent results have been published. Taken together, we conducted a meta-analysis to provide an up-to-date assessment and aimed to explore the relationship between elevated PCT and the severity of COVID-19.

## 2. Methods

The PRISMA (preferred reporting items for systematic reviews and meta-analyses) statement was used as a guideline for reporting this meta-analysis [13].

### 2.1. Search Strategy

Electronic databases (PubMed, Web of Science, and Embase) were systematically searched by two independent authors (Yue Shen and Cheng Cheng) from inception up to 2 April 2021. The search terms included PCT and COVID-19 (Appendix A). Considering that the first article of COVID-19 was published in Jan 2020, search filters were used to limit the results to the “2010–2021” publication time range and English articles. Additionally, we manually searched the reference lists in the relevant articles.

### 2.2. Inclusion Criteria and Study Selection

Two authors analyzed the titles, abstracts, and full texts of all these articles, and reviewed the retrieved studies using standardized data collection tables. In the case of any disagreement, a consensus should be reached by the third investigator.

The study was included if it: (1) reported the risk (including OR, RR, and HR) between PCT and the severity of COVID-19; (2) conducted research in general adults (≥18 years old); and (3) adjusted for at least one confounding factor. The patients of severe COVID-19 are those who met the criteria of either the WHO [14] or the National Health Commission of China [15] for severe and critical COVID-19, meeting any of the following criteria: (1) respiratory rate > 30 breaths/min; (2) oxygen saturation < 90% on room air; (3) severe respiratory distress (accessory muscle use, inability to complete full sentences); (4) critical complication (acute respiratory distress syndrome, sepsis, septic shock, or other conditions that would normally require the provision of life-sustaining therapies such as mechanical ventilation or vasopressor therapy); or (5) dead.

The study was excluded if it: (1) had a sample of fewer than 20 patients; (2) was a review article, meta-analysis, conference summary, clinical trial report, or grey literature (material is produced on all levels of government, academics, business and industry in print and electronic formats, but which is not controlled by commercial publishers) [16]; (3) was a case report or case series; (4) included special participants (e.g., cardiovascular, cancer or diabetes participants); (5) did not report on the results of severe, critical or dead patients; and (6) only reported the unadjusted effect estimates.

### 2.3. Data Extraction

The following information was extracted from eligible studies: name of the first author, country of publication, study design, sample size, research outcomes, participant characteristics (sex, age), adjusted effect estimate, 95% confidence interval (CI), and confounder(s) in the multivariable analysis.

### 2.4. Quality Assessment

The quality of eligible studies was evaluated by the Newcastle–Ottawa scale (NOS), which has eight items in three domains: subject selection (four scores); study group comparability (one score); and outcome(s) evaluation (three scores) [17]. Each study has a maximum score of eight. A study with a score of ≥6 was considered a high-quality study.

### 2.5. Statistical Methods

We performed random-effects meta-analyses [18] to pool the adjusted effect estimates. Two-side *p*-value < 0.05 was considered statistically significant. Q and *I*^2^ were used to assess heterogeneity between studies [19]. *I*^2^ ≤ 50% or *p* ≥ 0.1 demonstrated no significant heterogeneity, and a fixed-effects model was used. *I*^2^ > 5% or *p* < 0.1 indicated a significant heterogeneity, and a random-effects model was applied.

Subgroup analyses were conducted by sex, age, sample size, effect estimate, outcome, and adjusted confounders. Additionally, sensitivity analyses excluded one study at a time to assess the stability of the results. Egger’s test [20] was used to evaluate publication bias. Stata software (Version 11.2; Stata Corporation, College Station, TX, USA) was used for all statistical analyses.

## 3. Results

### 3.1. Literature Search and Study Characteristics

A total of 1805 studies were searched from electronic literature databases (Appendix A). We deleted 999 duplicate articles and 220 irrelevant articles according to the title and abstract. Then, the full texts of the remaining 274 articles were carefully reviewed. Of them, 264 were further excluded through exclusion criteria. Finally, 10 studies with 7716 patients were included in the meta-analysis [11,12,21,22,23,24,25,26,27,28].

The characteristics of the 10 included studies are shown in Table 1. Of these, six studies were conducted in China, two from the United States [22,26], one from Italy [27], and one from Turkey [12]. Only one study was a prospective cohort study [27], while the others were retrospective cohort studies. The outcome in nine studies was death, and the remaining one was ICU [24]. The detailed NOS scores are shown in Appendix A, and all studies were considered to be of high quality with an NOS score ≥ six scores.

### 3.2. PCT and the Severity of COVID-19

The forest plot, using a random-effects model, of the relation of PCT and the severity of COVID-19 risk presented that patients with elevated PCT on admission had a significantly increased risk of severe COVID-19 by 77% (pooled effect estimate: 1.77, 95% CI: 1.38–2.29) (Figure 1).

### 3.3. Subgroup and Sensitivity Analyses

Among the 10 studies, the primary outcome of nine studies was death, and the meta-analysis of the association with elevated PCT and the risk of death showed that elevated PCT was a significant risk factor for the mortality of COVID-19. It also had a significantly increased risk of the mortality of COVID-19 by 77% (pooled effect estimate: 1.77, 95% CI: 1.36–2.30, random-effects model) (Figure 2).

Subgroup analysis based on predetermined factors revealed heterogeneity in the comparison (Table 2). For elevated PCT, which displayed significantly increased risks for the severity of COVID-19 in the primary analyses, all subgroup analyses maintained the positive correlation (Appendix A). It indicated that the predetermined factors, such as sex, age, sample size, effect estimate, outcome, and adjusted confounders, might be risk factors. Among them, adjustment for diabetes reduced the heterogeneity. When studies were stratified by adjustments for diabetes, the summary estimate effect was weaker among the studies with such adjustments than among those without (RR = 1.23, 95% CI 1.06–1.44 vs. RR = 3.20, 95% CI 1.59–6.45). Sensitivity analysis (Figure 3) showed that the studies excluded one by one had no significant impact on the results. Egger’s test found no evidence of publication bias (*p* = 0.106) (Figure 4).

## 4. Discussion

After controlling the potential confounders, our meta-analysis revealed that elevated PCT on admission was positively associated with the severity of COVID-19 (pooled effect estimate: 1.77, 95% CI: 1.38–2.29), and the relationship also existed between elevated PCT on admission and dead patients (pooled effect estimate: 1.77, 95% CI: 1.36–2.30). Overall, the analyses involved good-to-high-quality studies with close to 7716 participants. The results were robust concerning the association between elevated PCT on admission and the severity of COVID-19; familiar results were also found in subgroup and sensitivity analyses.

Given the high heterogeneity of the included studies, various subgroup analyses were performed to identify the potential sources of heterogeneity. We found that the high heterogeneity may be partly explained by the confounding factor of diabetes, and heterogeneity lowered in studies that adjusted for diabetes compared to those that did not adjust for diabetes. Future studies adjusting for the relevant confounding factors need to explore the inherent association between PCT and the severity of COVID-19.

The serum PCT was a potential biomarker to evaluate and predict the severity of COVID-19 which was not influenced by corticosteroid treatment [29]. In the absence of infection, the transcription of the *Calc-I* gene is suppressed, and the calcitonin peptide families circulate as free peptides at low concentrations in the serum of healthy individuals. When systemic bacterial infection occurs, the tissue-specific control of *Calc-I* is disrupted, and the expression of *Calc-I* gene is increased, leading to the massive release of PCT [30]. Research has shown that the synthesis and secretion of PCT may be induced either directly via endotoxins as well as lipopolysaccharides or indirectly via pro-inflammatory cytokines, such as tumor necrosis factor-α and IL-6 [31]. PCT is formed by transcription and translation in C cells near thyroid follicles using *Calc-I* as the template. Under normal conditions, PCT is quickly cleaved into three parts: N-terminal fragment, peptides calcitonin and katacalcin [31]. SARS-CoV-2 can trigger an inflammatory cascade via the release of pro-inflammatory cytokines, such as IL-1β and IL-6, after activating Toll-like receptors which are also known to stimulate the release of PCT [32]. Patients with severe COVID-19 can develop immune hyperactivation and cytokine storm accompanied with a high level of PCT [33]. It has been shown that a reasonable explanation for elevated PCT in severe COVID-19 patients is co-infection with bacteria. Severe, critical and dead COVID-19 patients were more likely to have a co-infection or multiple organ failure [34]. In addition, a study has pointed out that severe and critical COVID-19 patients might show relatively significant immunosuppression [35], which increased the probability of combining bacterial infection to a certain extent. PCT is a precursor of calcitonin, which is secreted by neuroendocrine cells such as C cells of the thyroid and pancreatic tissues [36]. Bacterial infection would induce the expression of *Calc-I*, which encodes PCT, leading to a massive release of PCT [30]. Besides, many studies have demonstrated that PCT plays an important role in the diagnosis of systemic bacterial infection [36,37]. In severe infections (bacterial, fungal, and parasitic), sepsis and multiple organ failure, serum PCT levels are elevated in serum [36]. Elevated serum PCT levels have also been reported in patients with pulmonary infection, which also corroborates the above opinion [38]. Besides being a biomarker of severity, procalcitonin is a mediator of sepsis and possibly COVID-19. The latest research suggests that severe respiratory viral infection induced procalcitonin in the absence of bacterial pneumonia [39]. It upregulates surface markers on neutrophils/lymphocytes and upregulates cytokines and reactive oxygen species (ROS). This positive feedback between procalcitonin and proinflammatory cytokines subsequently culminates in a severe systemic inflammatory response [39]. However, our findings suggested that the COVID-19 patients with elevated PCT had a higher risk of severe COVID-19, hinting that PCT might be a potential biomarker to evaluate the severity of COVID-19 and predict the prognosis—further research should be conducted to verify the relationship.

To our knowledge, risk factors for the severity of COVID-19, including demographic factors, symptoms, comorbidities, complications, laboratory indicators, diet, and lifestyle, have been reviewed [40]. As previously stated, a bacterial infection is the main cause of acute exacerbation of chronic obstructive pulmonary disease (AECOPD) [41], and patients with AECOPD commonly have an elevated PCT level [42]. Likewise, elevated PCT is generally detected among patients with interstitial lung disease after bacterial infection [43]. Consequently, the above complications will increase the level of serum PCT in patients, affecting the association between PCT and the severity of COVID-19. To effectively control the impact of confounding factors, we performed a meta-analysis applying adjusted effect estimates.

There are several limitations in our meta-analysis. Firstly, noticeable heterogeneity exists in our study. Although sensitivity analysis manifested that our results were robust, we also need to control the magnitude of heterogeneity by selecting appropriate sample sizes and other methods. Secondly, the selected studies were mainly from China, and only two studies were from foreign countries. More studies are needed to confirm our results. Thirdly, most studies in our meta-analysis were retrospective, whose information is mainly from electronic medical records, so the lack of data and subjective information is inevitable. Finally, this study is underpowered to investigate the underlying mechanism of PCT with the severity of COVID-19, and the specific molecular mechanisms of the disease need to be further studied.

## 5. Conclusions

The present study provided evidence that elevated PCT is positively associated with an increased risk of the severity of COVID-19. PCT is a potential biomarker to evaluate the severity of COVID-19 and predict the prognosis.

## Figures and Tables

**Figure 1 medicina-57-00594-f001:**
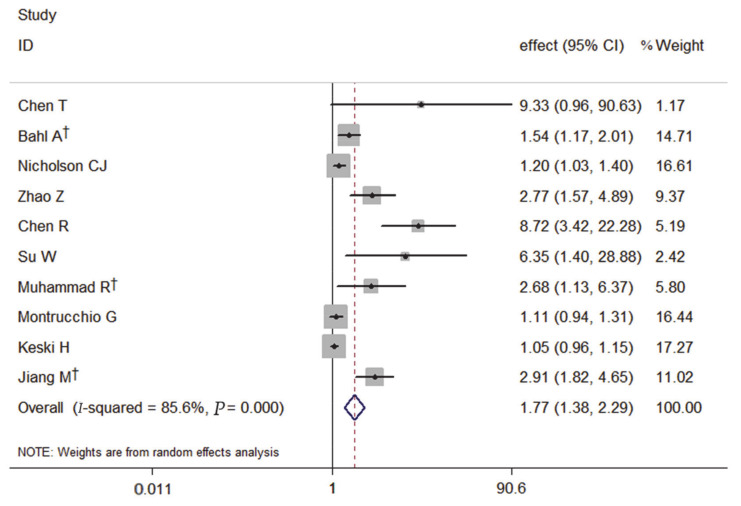
Forest plot of the meta-analysis of cohort studies on PCT and the severity of COVID-19 (CI, confidence interval). † indicates combined effects based on subgroups.

**Figure 2 medicina-57-00594-f002:**
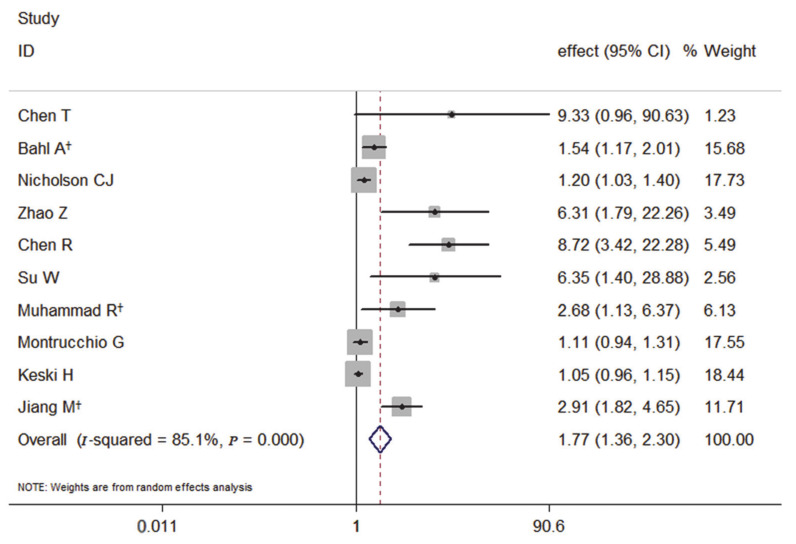
Forest plot of the meta-analysis of cohort studies on elevated PCT and the mortality of COVID-19 (CI, confidence interval). † indicates combined effects based on subgroups.

**Figure 3 medicina-57-00594-f003:**
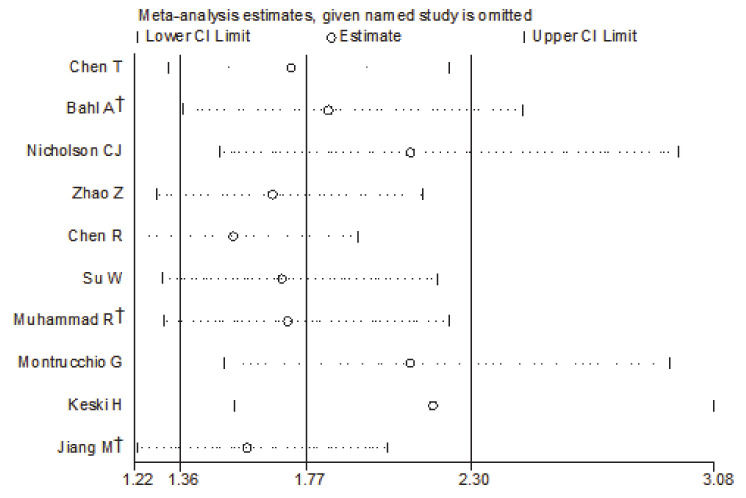
Influence analysis excluding one study at a time for the meta-analysis on PCT and the severity of COVID-19. † indicates combined effects based on subgroups.

**Figure 4 medicina-57-00594-f004:**
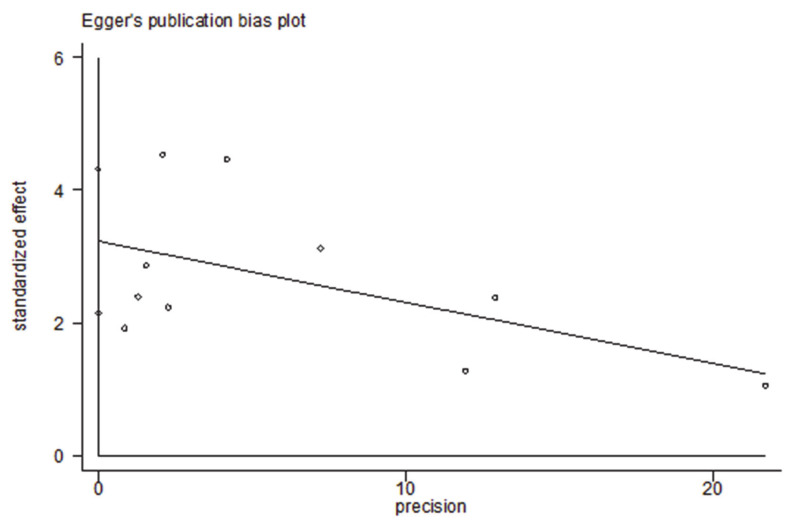
Egger’s test of the meta-analysis on PCT and severity of COVID-19 (*p* = 0.106).

**Table 1 medicina-57-00594-t001:** Characteristics of the included studies.

First Author	Country	Cases (*n*)	Age (years)	Male *n* (%)	Study Design	Outcomes	Adjusted OR/RR/HR (95% CI)	Confounders
Chen	China	55	NA	NA	R	death	OR 9.33 (0.96, 90.63)	Age, Comorbidities, Breath Shortness, Time From Illness Onset To First Hospital Admission, AST, Cr, LDH, CRP
Bahl	USA	1461	62 ± 17.8	770 (52.7)	R	death	HR 1.23 (0.78, 1.94)	Age, Sex, Race, BMI, CAD, Diabetes Mellitus, Hypertension, Respiratory Rate, Breaths Per Minute, Blood Oxygen Saturation, WBC, Hemoglobin, ALT, Creatinine, D-Dimer, Lactic Acid
							HR 1.37 (0.84, 2.23)	
							HR 2.11 (1.34, 3.31)	
Nicholson	China	1042	64 ± 16.3	592 (56.8)	R	death	OR 1.202 (1.033, 1.399)	Age, Sex, Diabetes Mellitus, Statin (Chronic Use), ALB, CRP, MCV, Neut: Lymph Ratio, PLT
Zhao	China	641	60	384 (59.9)	R	death	OR 6.31 (1.79, 22.26)	Age, Heart Failure, LDH, COPD, SpO_2_, Heart Rate
						ICU	OR2.77 (1.57, 4.89)	LDH, Smoking, SpO_2_, LYM count
Chen	China	1590	47.0 ± 65.2	904 (56.9)	R	death	HR 8.72 (3.42, 22.28)	Age, CHD, CVD, Dyspnea, AST, TBIL, Creatinine
Su	China	651	60.7 ± 16.3	332 (51.0)	R	death	OR 6.350 (1.396, 28.882)	Sex, Age, WBC, NEU, LYM Count, PLT Count, CD3, CD4, CD8
Muhammad	United States	200	58.9 ± 15.1	121 (60.5)	R	death	OR 2.68 (1.13, 6.37)	Age, Hypertension, CAD, Dyslipidemia, Chronic Kidney Disease, Stroke, Oxygen Saturation, Creatinine, BUN, CPK, Troponin, Lactic Acid, LDH, CRP, Initial D-Dimer, Ferritin, Highest D-Dimer
Montrucchio	Italy	57	63.0 ± 12.9	50 (87.7)	p	death	OR 1.113 (0.945, 1.312)	Age, Gender, CVD, Diabetes Mellitus, MR-proADM
Keski	Turkey	302	57.1 ± 17.6	148 (49.0)	R	death	HR 1.05 (0.96, 1.15)	Age, Hypertension, NLR, C-reactive protein, Ferritin, Prothrombin time, aPTT
Jiang	China	1717	61.3 ± 14.1	739 (48.17)	R	death	HR 2.91 (1.82, 4.65)	Age, Gender, COPD, AST, hs-CRP, hs-TnI, WBC, LYM count, D-dimer

Note: The values of age are mean ± standard deviation (SD) or median (interquartile range, IQR); the values of male are n (%). Abbreviation: ALB, albumin; ALT, alanine aminotransferase; aPTT, activated thromboplastin time; AST, aspartate aminotransferase; BMI, body mass index; BUN, blood urea nitrogen; CAD, coronary artery disease; CHD, coronary heart disease; CVD, cardiovascular disease; COPD, chronic obstructive pulmonary disease; COVID-19, coronavirus disease 2019; CI, confidence interval; CRP, C-reactive protein; hs-CRP, high-sensitivity C-reactive protein; hs-TnI, high-sensitivity troponin I; HR, hazard ratio; LDH, lactate dehydrogenase; LYM, lymphocyte; MCV, mean corpuscular volume; MR-proADM, mid-regional pro-adrenomedullin; NEU, neutrophil; OR, odds ratio; PCT, procalcitonin; P, prospective study; PLT, platelet; R, retrospective study; SpO_2_, decreased pulse oxygen saturation; TBIL, total bilirubin; WBC, white blood cell count.

**Table 2 medicina-57-00594-t002:** Subgroup analysis on the association between PCT and the severity of COVID-19.

Subgroup	Number of Study	Pooled Effects (95% CI)	*I*^2^ (%)	*P*
All studies	10	1.77 (1.38–2.29)	85.6	0.000
Effect estimate				
OR	6	1.65 (1.18–2.31)	75.0	0.001
HR	4	2.16 (1.21–3.84)	92.7	0.000
Sex (male, %)				
≥55	5	1.90 (1.28–2.83)	86.3	0.000
<55	4	1.81 (1.09–3.00)	89.4	0.000
NA	1	9.33 (0.96–90.63)		
Sample size				
≥500	6	2.47 (1.55–3.94)	86.6	0.000
<500	4	1.16 (0.93–1.45)	63.4	0.042
Outcomes				
Death	9	1.66 (1.29–2.14)	85.1	0.000
others	1	2.77 (1.57–4.89)		
Age				
≥60	6	1.68 (1.25–2.25)	82.4	0.000
<60	3	2.73 (0.75–9.88)	91.6	0.000
NA	1	9.33 (0.96–90.63)		
Diabetes				
1	3	1.23 (1.06–1.44)	50.7	0.132
0	7	3.20 (1.59–6.45)	89.6	0.000
Hypertension				
1	3	1.40 (0.94–2.08)	81.9	0.004
0	7	2.31 (1.52–3.51)	86.8	0.000

## Data Availability

The data that support the findings of this study are available from the corresponding author upon reasonable request.

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
