# Peer review of "Elevated Procalcitonin Is Positively Associated with the Severity of COVID-19: A Meta-Analysis Based on 10 Cohort Studies"

_medicina, 2021, doi:10.3390/medicina57060594_

Round 1
Reviewer 1 Report
Overview
This manuscript described a systematic review of the literature in an attempt to better describe the association between severity of COVID-19 infection and Procalcitonin levels.
Major
It is not clear what is meant by ‘severity’ in the abstract. How is this being measured? The main text refers to the WHO document – I think table 2 in this document needs to be described in further detail or ideally reproduced in order to make the definition clear to the readership
Procalcitonin is a marker of systemic inflammatory activity, but there is no mention of it’s close assscoiation with bacterial infection specifically.
Line 94 states a two sided p-value of <0.1 was significant. Why was this chosen? It is an unusual cut off value.
I would highly recommend a statistical editor or reviewer with statistical expertise should analyse the manuscript
Your forest plot could do with being larger in a separate figure – it is the main point of your study. It is not clear what is being measured. Is there a cut off for PCT used? What does ‘elevated PCT’ mean – bearing in mind there is no definitive cut off value?
In your discussion, it is mentioned that COPD and ILD increase PCT levels. This is not true in the absence of bacterial infection and requires further explanation.
I do not agree with your title! The most likely mechanism of PCT elevation in COVID-19 is due to bacterial co-infection (as you eventually point out in the discussion). Thus, elevated PCT does no increase the risk of severity, you have possibly demonstrated only that it is correlated with an increased risk of severe outcomes including death. The way your data are presented needs re-writing to appropriately represent this.
You appear to have adjusted your analyses for known risk factors separately in the supplementary materials, but surely the most interesting outcome would be to see if PCT predicts seere outcomes after adjustment for ALL of these other factors in a full multivariate analysis. I think if this is to be genuinely additive to the current literature, a multivariate model is required. I re-iterate my recommendation for a statistical review here.
The written English is of a lower standard than would be expected of a final published manuscript and would benefit from being reviewed by someone with lingual proficiency in English.
Minor
It is mentioned in the introduction that IL6 may be a target for COVID-19 treatment. It seems a little more certain than that. Tocilizumab has been shown to reduce mortality in the RECVOERY trial and should be mentioned here.
Please define what is meant by ‘grey literature’ (line 81)
Author Response
Response to Reviewer Comments
Major
Point 1 It is not clear what is meant by ‘severity’ in the abstract. How is this being measured? The main text refers to the WHO document – I think table 2 in this document needs to be described in further detail or ideally reproduced in order to make the definition clear to the readership
Response 1:
Thanks for your reviewer comments. The severity of COVID-19 has been defined in COVID-19 Clinical management: living guidance, the patients of severe COVID-19 meet any of the following criteria: (1) respiratory rate > 30 breaths/min; (2) oxygen saturation < 90% on room air; (3) severe respiratory distress (accessory muscle use, inability to complete full sentences); (4) critical complication(acute respiratory distress syndrome, sepsis, septic shock, or other conditions that would normally require the provision of life-sustaining therapies such as mechanical ventilation or vasopressor therapy); or (5) dead. We have described the definition of ‘severity of COVID-19’ in Method part in lines 93 - 98.
Point 2 Procalcitonin is a marker of systemic inflammatory activity, but there is no mention of its close association with bacterial infection specifically.
Response 2:
Preliminary reports indicated that PCT was mainly induced during severe systemic inflammation caused by bacterial infections. In severe bacterial infection, the increase of PCT in serum was significantly higher than that of other pathogenic (fungi, viruses, or other atypical microorganisms). Furthermore, PCT might be an indicator of bacterial infection, especially systemic bacterial infection. LPS in the cytoderm of gram-negative bacterium could activate macrophages to release cytokines such as IL-1, TNF-α and IL-6, thereby inducing the increase of PCT. Additionally, the gram-positive bacterium could produce toxic shock syndrome toxin-1, which not only induced monocytes to produce IL-1 and TNF but also activated massive T-cells and monocytes to release cytokines as superantigens. Both of them could elevate the level of PCT. However, LSP, the main pathogenic substance of gram-negative bacterium, could directly activate macrophages and induce the production of PCT. The main pathogenic substance of gram-positive bacterium was enterotoxin, which had a weak ability to stimulate the production of PCT. (Reference: Opal SM, Cohen J. Clinical gram-positive sepsis: does it fundamentally differ from gram-negative bacterial sepsis? Crit Care Med. 1999;27(8):1608-16.)
Point 3 Line 94 states a two-sided p-value of <0.1 was significant. Why was this chosen? It is an unusual cut-off value.
Response 3:
In meta-analysis, the different cut-off values of p-value were commonly selected for diverse purposes. For example, Pan et al. regarded a two-sided p-value of < 0.05 as a statistically significant result for pooled effect estimates (Pan H, Hibino M, Kobeissi E, et al. Blood pressure, hypertension and the risk of sudden cardiac death: a systematic review and meta-analysis of cohort studies. European journal of epidemiology. 2020;35(5):443-54.). And, in analyzing the statistical heterogeneity between studies, the p-value of < 0.10 was defined as indicating significant heterogeneity. The same cut-off values of p-value were also used in the Yu’s study(Yu HH, Qin C, Chen M, et al. D-dimer level is associated with the severity of COVID-19. Thromb Res. 2020;195:219-25.). Consequently, we have made a clearer illustration of the cut-off value in Method part in lines 119 - 122.
Point 4 I would highly recommend a statistical editor or reviewer with statistical expertise should analyze the manuscript
Response 4:
We have invited statistical experts Jing Xie (Email: Xiejing1972@hotmail.com) to conduct quality control of the study.
Point 5 Your forest plot could do with being larger in a separate figure – it is the main point of your study. It is not clear what is being measured. Is there a cut-off for PCT used? What does ‘elevated PCT’ mean – bearing in mind there is no definitive cut-off value?
Response 5:
We have presented the figures separately at the end of the article to ensure that it is clear to read. All patients underwent the assessment of clinical and routine laboratory tests, typically within 24 hours of admission, which was the first test results available. Descriptive data were presented as mean ± standard deviation or median and interquartile ranges of 25th and 75th percentiles. Subsequently, multivariable models adjusting for confounders to predict the relationship between PCT and the severity of COVID-19. To our knowledge, the cut-off values of PCT are inconsistent in studies. Hence, we compared the high level of PCT with the low level of PCT to estimate the relationship. Therefore, considering the above reasons, 'elevated PCT' was used in the article.
Point 6 In your discussion, it is mentioned that COPD and ILD increase PCT levels. This is not true in the absence of bacterial infection and requires further explanation.
Response 6:
We have further explained the elevated PCT in COPD and ILD patients mentioned in the Discussion part in lines 222 - 226, increasing the premise of bacterial infection. Specific contents are as follows: “As previously stated, a bacterial infection is the main cause of acute exacerbation of the chronic obstructive pulmonary disease, and patients with COPD commonly have an elevated PCT level. Likewise, elevated PCT is generally detected among patients with interstitial lung disease after bacterial infection. Consequently, the above complications will increase the level of serum PCT in patients, affecting the association between PCT and the severity of COVID-19.”
Point 7 I do not agree with your title! The most likely mechanism of PCT elevation in COVID-19 is due to bacterial co-infection (as you eventually point out in the discussion). Thus, elevated PCT does not increase the risk of severity, you have possibly demonstrated only that it is correlated with an increased risk of severe outcomes including death. The way your data are presented needs re-writing to appropriately represent this.
Response 7:
We have reworked our title and the relevant expressions in the main text and supplementary. The modified title is Elevated procalcitonin is positively associated with the severity of COVID-19: A meta-analysis based on 10 cohort studies. Besides being a biomarker of severity, procalcitonin is a mediator of sepsis and possibly COVID-19. It upregulates surface markers on neutrophils/lymphocytes and upregulates cytokines and reactive oxygen species (ROS). This positive feedback between procalcitonin and the pro-inflammatory cytokines subsequently culminates in a severe systemic inflammatory response (Reference: Gautam S, Cohen AJ, Stahl Y, et al. Severe respiratory viral infection induces procalcitonin in the absence of bacterial pneumonia. Thorax. 2020;75(11):974-8.). However, the precise molecular mechanism of PCT elevation in COVID-19 was also not very clear. Our findings suggested the COVID-19 patients with elevated PCT had a higher risk of the severity of COVID-19, hinting PCT might be a potential biomarker to evaluate the severity of COVID-19 and predict the prognosis, further research should be conducted to verity the relationship.
Point 8 You appear to have adjusted your analyses for known risk factors separately in the supplementary materials, but surely the most interesting outcome would be to see if PCT predicts severe outcomes after adjustment for all of these other factors in a full multivariate analysis. I think if this is to be genuinely additive to the current literature, a multivariate model is required. I re-iterate my recommendation for a statistical review here.
Response 8:
In the present study, we extracted the adjusted estimates from the eligible studies, which used the multivariate analysis adjusted for confounders. So, our meta-analysis provided the adjusted pooled effect estimates of the relationship between PCT and the severity of COVID-19 and found a positive relationship, as shown in Figure 1 of the main text. Then, as the general meta-analysis, we conducted subgroup analysis on some characteristic factors included in the articles to analyze the heterogeneity between studies (Reference: Wei Q, Lin H, Wei RG, et al. Tocilizumab treatment for COVID-19 patients: a systematic review and meta-analysis. Infect Dis Poverty. 2021;10(1):71.). Meanwhile, we also performed sensitivity analysis and publication bias analysis to evaluate the robustness of the articles.
Point 9 The written English is of a lower standard than would be expected of a final published manuscript and would benefit from being reviewed by someone with lingual proficiency in English.
Response 9:
We have revised the manuscript carefully. We believe that the language is now acceptable for the review process. If you have any other suggestions please point them out, and thank you for reviewing this article.
Minor
Point 1 It is mentioned in the introduction that IL6 may be a target for COVID-19 treatment. It seems a little more certain that Tocilizumab has been shown to reduce mortality in the RECOVERY trial and should be mentioned here.
Response 1:
Tocilizumab, an IL-6 receptor blocker, neutralises IL-6 activity by binding the IL-6R at the IL-6-binding epitope. The preliminary results from the RECOVERY trial identified a slight decrease in 28-day mortality in patients treated with the combination of steroid and tocilizumab versus steroid alone. Furthermore, mortality was 4–5% higher in patients who received tocilizumab monotherapy compared with those who did not receive tocilizumab or steroids (Reference: Azithromycin in patients admitted to hospital with COVID-19 (RECOVERY): a randomised, controlled, open-label, platform trial. Lancet (London, England). 2021;397(10274):605-12.). We have added the relevant discussion in the Introduction part in lines 49 - 55.
Point 2 Please define what is meant by ‘grey literature’ (line 81)
Response 2:
Forth International Conference on Grey Literature: New Frontiers in Grey Literature had defined grey literature as follows, which was produced on all levels of government, academics, business and industry in print and electronic formats, but which is not controlled by commercial publishers. We have attached the definition in the Method part in lines 101 - 103.

Reviewer 2 Report
Major comments
- Is there any association of procalcitonin and complement activation pathway that linked to severity of Covid19? I have not seen any references on this.
- How procalcitonin regulates inflammation and what are the controlling factors for it?
- It will be good to provide the transcription factors that regulate the expression of Procalcitonin in Covid19
Author Response
Response to Reviewer 2 Comments
Major
Point 1 Is there any association of procalcitonin and complement activation pathway that linked to the severity of Covid19? I have not seen any references on this.
Response 1:
We have reworked the discussion about the potential mechanisms of correlations between PCT and the severity of COVID-19 in the third paragraph of the Discussion part. The contents as follows, “The serum PCT was a potential biomarker to evaluate and predict the severity of COVID-19 which was not influenced by corticosteroid treatment. In the absence of infection, transcription of the Calc-I gene is suppressed, and the calcitonin peptide families circulate as free peptides at low concentrations in the serum of healthy individuals. When systemic bacterial infection occurs, the tissue-specific control of Calc-I is disrupted, and the expression of Calc-I gene is increased, leading to the massive release of PCT. Research has shown that the synthesis and secretion of PCT may be induced either directly via endotoxin as well as lipopolysaccharide or indirectly via pro-inflammatory cytokines, such as tumor necrosis factor-α and IL-6. PCT is formed by transcription and translation in C cells near thyroid follicles using Calc-I as the template.” There were not seen any researches on the association of procalcitonin and complement activation pathway that linked to the severity of Covid19. Further research should be performed to understand the complex processes by which PCT signaling occurs is crucial for the correct interpretation of PCT concentrations in the serum, the use of PCT as a critical care biomarker. We sincerely appreciate your help in the point out our mistakes.
Point 2 How procalcitonin regulates inflammation and what are the controlling factors for it?
Response 2:
Previously, Nylen et al. suggested a role of PCT as a mediator of inflammation. In a minimal-mortality hamster model, these authors demonstrated that injection of PCT augmented mortality from sepsis and that neutralizing antibodies reversed the effect (Nylen E. Mortality is increased by procalcitonin and decreased by an antiserum reactive to procalcitonin in experimental sepsis. [J]. Critical Care Medicine, 1998, 26(6):1001.). The reported sequence homologies between PCT and other human cytokines, such as TNF-a family, IL6, G-CSF and MGF family, etc., support this hypothesis. However, its specific mechanism still was not clear, further possible functions of PCT need to be studied. In the absence of infection, transcription of the Calc-I gene is suppressed, and the calcitonin peptide families circulate as free peptides at low concentrations in the serum of healthy individuals. When systemic bacterial infection occurs, the tissue-specific control of Calc-I is disrupted, and the expression of Calc-I gene is increased, leading to the massive release of PCT.
Point 3 It will be good to provide the transcription factors that regulate the expression of Procalcitonin in Covid-19.
Response 3:
When systemic bacterial infection occurs, the tissue-specific control of Calc-I is disrupted, and the expression of Calc-I gene is increased, leading to the massive release of PCT. Therefore, Calc-I gene might participate in the regulation of the expression of PCT in COVID-19. Importantly, Gautam found that PCT expression not only persisted despite IFN signaling in severe respiratory viral infection, but it also correlated positively with IFN-γ (Gautam S, Cohen AJ, Stahl Y, et al. Severe respiratory viral infection induces procalcitonin in the absence of bacterial pneumonia. Thorax. 2020;75(11):974-8.). At present, the precise regulatory mechanism of PCT and COVID-19 was unclear. An adequate understanding of the complex processes by which PCT signaling occurs is crucial for the correct interpretation of PCT concentrations in the serum, the use of PCT as a critical care biomarker.

Round 2
Reviewer 1 Report
My comments have been appropriately addressed. The manuscript is ready for publication.